# Incidence of Anaplastic Large Cell Lymphoma and Breast-Implant-Associated Lymphoma—An Analysis of a Certified Tumor Registry over 17 Years

**DOI:** 10.3390/jcm9051247

**Published:** 2020-04-25

**Authors:** Lukas Prantl, Michael Gerken, Florian Zeman, Michael Leitzmann, Michael Koller, Monika Klinkhammer-Schalke, Matthias Evert, Britta Kuehlmann, Niklas Biermann

**Affiliations:** 1University Center of Plastic, Hand- and Reconstructive Surgery, University Hospital Regensburg, Franz-Josef-Strauß-Allee 11, 93053 Regensburg, Germany; britta.kuehlmann@ukr.de (B.K.); niklas.biermann@ukr.de (N.B.); 2Tumor Center Regensburg, Institute for Quality Assurance and Health Services Research, University Hospital Regensburg, Franz-Josef-Strauß-Allee 11, 93053 Regensburg, Germany; michael.gerken@ukr.de (M.G.); monika.klinkhammer-schalke@ukr.de (M.K.-S.); 3Center for Clinical Studies, University Hospital Regensburg, Franz-Josef-Strauß-Allee 11, 93053 Regensburg, Germany; florian.zeman@ukr.de (F.Z.); michael.koller@ukr.de (M.K.); 4Department of Epidemiology and Preventive Medicine, University Hospital Regensburg, Franz-Josef-Strauß-Allee 11, 93053 Regensburg, Germany; michael.leitzmann@ukr.de; 5Department of Pathology, University Hospital Regensburg, Franz-Josef-Strauß-Allee 11, 93053 Regensburg, Germany; matthias.evert@ukr.de; 6Division of Plastic and Reconstructive Surgery, Department of Surgery, Stanford University, 770 Welch Road, Palo Alto, CA 94304, USA

**Keywords:** anaplastic large cell lymphoma, CD-30 positive, ALK-negative, breast implants, incidence

## Abstract

Background: Breast-implant-associated anaplastic large cell lymphoma (BI-ALCL) and primary breast ALCL are rare extranodal manifestations of non-Hodgkin lymphoma. The rarity of both diseases, along with unreleased sales data on breast implants and constant updates of classification systems impede the calculation of an exact incidence. Methods: The database of the Tumor Center Regensburg in Bavaria was searched for patients with CD30-positive and ALK-negative anaplastic large cell lymphoma between 2002 and 2018. These lymphomas were identified by the ICD-O-3 morphology code “97023” and were cross-checked by searching the diagnosis by name the and ICD-10 code C84.7. Furthermore, we tried to calculate the incidence rates and corresponding 95% confidence intervals, standardized to 1,000,000 implant years of breast-implant-associated anaplastic large cell lymphoma and primary breast anaplastic large cell lymphoma. Results: Twelve ALK-negative and CD30-positive anaplastic large cell lymphomas were identified out of 170,405 malignancies. No case was found within the breast tissue and none of the patients had a previous history of breast implant placement. In five cases, lymph node involvement in close proximity to the breast was observed. Conclusion: We found a low incidence of anaplastic large cell lymphoma and no association to breast implants in these patients. A review of the current literature revealed inconsistent use of classification systems for anaplastic large cell lymphomas and potential overestimation of cases.

## 1. Introduction

Silicone is the most common material among implantable biomedical devices. In particular, silicone implants are used in surgery for breast augmentation and breast reconstruction. Since their introduction in the 1960s, their safety has been under debate [1], with the issue of silicone- and other filler-material-induced malignancies remaining controversial [2]. Breast-implant-associated anaplastic large cell lymphoma (BI-ALCL) is a rare type of non-Hodgkin lymphoma first described by Keech et al. [3] Among several histologic subtypes, the expression of CD30 (CD30-positive) and an absent receptor tyrosine kinase, the anaplastic lymphoma kinase (ALK-negative), are important characteristics of BI-ALCL [4,5]. Since 1997, approximately 800 cases have been reported worldwide, only twenty-four of which were listed by the German Federal Institute for Drugs and Medical Devices (BfArM) [6,7,8,9,10,11,12]. As known from other malignancies and given the rarity of the disease along with unreleased implant sales data, calculating an exact incidence is challenging [13,14]. Figures vary between 1:700 and 1:1,000,000 among women receiving breast implants [15,16].

The intent of our study was to determine the number of primary breast ALCL (PBL-ALCL) and BI-ALCL cases in a well-defined population and to calculate an occurrence rate based on histologically confirmed cases.

## 2. Materials and Methods

### 2.1. Study Design

This is a retrospective registry study to investigate the incidence of anaplastic large cell lymphoma and primary breast anaplastic large cell lymphoma and its association with breast implants.

### 2.2. Data Base and Data Extraction

The Tumor Center Regensburg was founded in 1991 and, according to estimates of the German Robert-Koch Institute (RKI), since 2002 has captured over 90% of all cancer cases occurring among the 2.1 million inhabitants of the districts “Upper Palatinate” and “Lower Bavaria”. This cancer registry collects epidemiological and clinical data from all patients with malignancies diagnosed and treated by specialists, general practitioners and clinicians in the region. The University Hospital Regensburg, over 50 regional hospitals and more than 1000 practicing doctors participated in this cross-sectorial documentation of cancer patients. The registry receives medical information from all pathologists and clinicians regarding dates of diagnosis, treatment, and follow-up care according to German Cancer Registry Laws. Physicians enter the data into case forms using computer-assisted documentation or send medical reports to the registry, where the data are extracted, recorded, and merged and entered into a central database by trained medical documentation specialists. No re-coding of the diagnosis is done, only a simple translation into the registry from the original pathologic report. Tumor recurrences and vital status are ascertained using clinical reports, residential information from population registries, and death certificates from local public health departments. Data are processed and secured according to the Bavarian Law of Cancer Registries. In rare cases and for special questioning, direct patient contact by phone is possible.

The baseline cohort in this study comprises patients with ICD-10-GM diagnoses of malignant neoplasms C00 to C97, excluding non-melanoma cancer of the skin (C44). According to our search algorithm, patients with ALK-negative and CD30-positive ALCL were identified by the ICD-O-3 morphology code “97023” and were cross-checked by searching the diagnosis by name and ICD-10 code C84.7. After removal of duplicates, all patient records were screened for eligibility. The complete medical history, including radiologic diagnostics, was then analyzed for previous breast surgeries, including implants. For confirmation, patients were contacted by phone. ALCLs other than CD30-positive and ALK-negative were identified by the codes 96373, 97143, 97183, 97253 and 97263 and were cross-checked for correct classification.

The pathologic diagnosis was again confirmed by secondary re-analysis of the pathology specimen by an independent pathologist.

We further identified the most recent and frequently cited publications, providing sufficient evidence to calculate a comparable incidence rate.

Data search and analysis within the clinical cancer registry was conducted in accordance with the Declaration of Helsinki, and the protocol was approved by the Ethics Committee of the University Regensburg (approval no. 15-170-0000).

### 2.3. Statistical Analysis

BI-ALCL incidence rates and corresponding 95% confidence intervals were calculated for the Regensburg data as the quotient of the number of BI-ALCL cases divided by the number of implant years within a specific time period, standardized to 1,000,000 implant years. Implant years were defined as follows:Implant years = Total Number of women × proportion of implants × observation period

Example: 100,000 women are living in the area of interest. Four percent of these women have breast implants and are observed over a period of 10 years. This results in a total number of implant years of 100,000 × 0.04 × 10 = 40,000. This means a period of 40,000 implant years is the denominator for the incidence rate. If 2 BI-ALCLs are observed within this period of 10 years, the incidence rate is 2 per 40,000 implant years. Standardizing to 1,000,000 implant years results in 50 BI-ALCL in 1,000,000 implant years.

In the same manner, incidence rates were calculated for five comparable and relevant publications in the field [10,15,17,18,19]. In some cases, missing information on certain parameters required us to make certain assumptions. Details can be found in the Results section.

## 3. Results

### 3.1. Descriptive Data

Between 2002 and 2018, a total of twelve ALK-negative and CD30-positive ALCL were identified by our cancer registry (Table 1).

Table 1: Annual numbers of ALCL cases identified by the codes 97023 96373, 97143, 97183, 97253 and 97263. Peripheral T-cell-lymphomas (PTCL) other than ALK-negative and CD-30 positive anaplastic large cell lymphoma were ALK-positive anaplastic large cell lymphoma (*N* = 56), anaplastic large cell lymphoma (T and null cell types) (*N* = 12), primary cutaneous anaplastic large cell lymphoma (*N* = 17), anaplastic large cell lymphoma without T- and B-cell markers (*N* = 4) and anaplastic large cell lymphoma not specified (*N* = 11). Except for the anaplastic large cell lymphomas without T- and B-cell markers (97236) and the anaplastic large cell lymphoma not specified (97253), ICD-O-3 and WHO-classification codes were equal at the time theses codes were used for classification.

To further describe all other lymphomas and primary malignancies solely in the breast, we included an organ-based (breast) search algorithm.

Table 2: We found a total of 25918 breast malignancies, with 25,897 cases of primary breast cancer. Among 5181 non-Hodgkin-lymphomas (ICD-10 C82–85) diagnosed between 2002 and 2018, 21 patients proved to have a localization in the breast (ICD-O-3 C50). None of these 21 patients showed a T-cell-lymphoma; all patients had B-cell-lymphomas (C82 follicular lymphoma *N* = 4, C83 non-follicular lymphoma *N* = 15, C85.9 non-Hodgkin lymphoma, unspecified *N* = 2).

Figure 1 represents the flow of patient identification according to the EQUATOR reporting guidelines (http://www.equator-network.org/reporting-guidelines/) [20].

The gender distribution was even, with six male and six female patients. Twelve cases within a 17-year observation period represent 0.7 cases per year and an average of 0.007% of all 170,394 malignancies except non-melanoma skin cancer (C44).

The mean age of diagnosis was 58.2 years (range, 18.2 to 82.4 years).

There was no case of BI-ALCL or primary breast ALCL, but immunohistochemically similar ALCL that was nodal in nature in twelve patients. One case included a neoplasm around the eyelid, two cases were found in the cervical lymph nodes, one case in the thoracic spine, one case in an abdominal lymph node and seven cases in multiple or disseminated lymph nodes. Of the disseminated cases, two included cervical and axillary lymph nodes, one case included the infra- and supraclavicular lymph nodes with infiltration of the cervical plexus, another two cases were detected in the cervical, supraclavicular and mediastinal lymph nodes, one case was found within the axillary and hepatic lymph nodes, and one final case included the axillary, inguinal, iliac and para-aortic lymph nodes.

Radiographic imaging, including thoracic CT scans, were available in all cases. During the same time period, 100 patients with ALCLs other than ALK-negative and CD30-positive histologic subtypes were identified, representing 0.05% of all malignancies except non-melanoma skin cancers (C44). These were peripheral T-cell lymphomas, including ALK-positive anaplastic large cell lymphoma, anaplastic large cell lymphomas (T and null cell types), primary cutaneous anaplastic large cell lymphoma and anaplastic large cell lymphomas not specified and without T- and B-cell markers. ICD-0-3 and WHO classification codes were equal for the ALK-positive, primary cutaneous and the T and null cell type anaplastic large cell lymphomas.

### 3.2. Incidence in the Regensburg Data Set

In our study, there was no case of a BI-ALCL between 2002 and 2018. According to the Bavarian Federal Statistical Office, an average of 900,000 women live in our region, and an estimated prevalence of breast implants according to de Boer of 3.3% yields a total of 504,900 implant years [15,21]. This allows calculation of an incidence rate of 0 (95%-CI: 0–7.31) per 1,000,000 implant years.

### 3.3. Incidence in Comparison Studies

De Boer et al. reported 32 cases of BI-ALCL between 1990 and 2016. The authors estimated the prevalence of women with breast implants at 3.3% according to an evaluation of chest radiographs but provided no information about the number of women in the respective time period. According to the German Federal Statistical Office (StBA), the Netherlands had an average of about 6.4 million women, aged between 20 and 70 years (age interval analogue to the BI-ALCL cases) per year [22]. This results in about 3,590,400 implant years, yielding an incidence rate of 8.9 (95%-CI: 6.1–12.6) per 1,000,000 implant years [15].

Wang et al. reported two cases of BI-ALCL between 1995–2012 among 2990 women with breast implants. This represents 44,394 implant years and an incidence rate of 45.1 (95%-CI: 5.6–162) per 1,000,000 implant years [18].

Largent et al. reported three BI-ALCL cases between 1996 and 2007 among 89,382 female patients with breast implants [19]. As published by the authors, this represents 204,682 implant years and an incidence rate of 14.6 (95%-CI: 3–42.7) per 1,000,000 implant years.

Doren et al. identified 100 BI-ALCL cases in the U.S. between 1996 and 2015 and calculated an incidence rate of 2.03 per 1,000,000 implant years. This results in a total of 49,261,084 implant years and a corresponding 95%-CI of 1.7–2.5 per 1,000,000 implant years [10].

Loch-Wilkinson et al. identified 38 BI-ALCL cases in total (cases were divided into three different implant types) between 1999 and 2015. Using the reported incidence rates of the implant types gives a total of 1,345,563 implant years and represents an incidence rate of 28.2 (95%-CI: 20–38.8) per 1,000,000 implant years [23].

The incidence rates and confidence intervals of the five studies are summarized in a Forest plot (Figure 2).

## 4. Discussion

The incidence and prevalence of primary ALCL of the breast and BI-ALCL show wide variation. Different classification models for hematopoietic malignancies, such as the World Health Organization (WHO)’s lymphoma classification and the International Classification of Diseases (ICD) and ICD for oncology (ICD-O), as well as inconsistent immunohistochemical analysis of cancer specimens, contribute to this heterogeneity.

This has led to controversy with regards to the use of breast implants and especially with textured surfaces [24,25,26,27]. The rarity of the disease along with insufficient data on women with breast implants and breast implant sales contributes to insufficient statistical information. Patients are usually identified by their ICD-O morphology classification code within a database.

In the database of the Tumor Center Regensburg, we detected twelve cases of CD30-positive and ALK-negative ALCL outside the breast tissue between 2002 and 2018. Any previous history of breast augmentation was safely excluded, leading to the assumption that no case of a BI-ALCL was detected. Most likely, these twelve cases do not represent primary ALCL of the breast but, rather, systemic presentations of ALK-negative and CD30-positive ALCL. However, in five cases, lymph node involvement in close proximity to the breast was observed. One female case involved the infraclavicular lymph nodes outside the thoracic wall, displaying the importance of precise workup.

These findings are supported by our previous work investigating 296 breast capsules between 2000 and 2015 during revision surgery. No CD30-positive and ALK-negative cells could be detected upon immunohistochemical analyses [28].

A few other studies have investigated the incidence of primary breast ALCL [11,29,30,31]. Based on the National Cancer Institute´s Surveillance Epidemiology and Results (SEER) program, Altekruse et al. reported an incidence rate of 3 per 100 million per year (*p* < 0.05). From 2011 onward, a steadily increasing incidence rate from 1 to 3 per 1 million person-years was suspected to result from over-reporting and increasing awareness [10,32].

Thomas et al. reported on primary breast lymphoma in the United States between 2000 and 2013 and found a total of 22 cases of primary breast ALCL, yielding an incidence rate of 0.037 per million women [29]. Again, the SEER database was used for analysis. However, the number of female patients with breast implants was not assessed by the SEER program, relativizing the incidence rate for BI-ALCL and PBL-ALCL.

In 2012, Largent et al. investigated American female patients with breast implants participating in six Allergan-sponsored studies and found an incidence rate of 1.46 (0.3–4.3) per 100,000 person years [19]. The control group was based on the SEER program and showed annual incidences of 4.28 (5.05–3.51) and 3.88 (4.58–3.19) per 100 million females aged older than 15 and 20 years, respectively. However, it was unknown whether the patients identified by the SEER investigation had a previous history of breast augmentation with implants. Furthermore, all three of the detected lymphomas during the clinical studies had a coexisting breast cancer. According to the classification models, two different systems were used. The SEER program reported cases of breast ALCL through the ICD-O Third Edition by the code “9714/3”. During the time of classification, “9714/3” coded for anaplastic large cell lymphoma expressing the lymphoma kinase (ALK-positive) and thus describes a different entity than BI-ALCL. The Allergan-sponsored studies classified lymphomas as either non-Hodgkin lymphoma or Hodgkin lymphoma, making a comparison challenging.

In 2016, Wang et al. investigated 123,392 Californian women, 2990 of whom reported having breast implants [18]. After an average follow-up of 14 years, only two out of ten women diagnosed with incident ALCL reported having breast implants. All patients diagnosed with ALCL were followed through linkages with the California Cancer Registry and were identified by the classification code “9714/3” according to the ICD-O Third Edition.

Until the first revision of the ICD-O Third Edition in 2013, ALCL with negative ALK-receptor status was not classified as an own entity and could not be selectively identified by database research. Thus, there may have been an overestimation of cases with BI-ALCL.

As to the WHO´s lymphoma classification, ALK-negative ALCL was included as a provisional entity in the 2008 edition by the code “9702/3”. In the third edition before 2008, ALK-negative and ALK-positive lymphomas were listed together as the same entity and were indistinguishable by their classification code alone. In the most recent edition of the classification, BI-ALCL is listed as a provisional entity with the same classification code as ALK-negative and CD30 positive ALCL (“9715/3”) [5]. So far, these two entities are difficult to distinguish by immunohistochemical or genetic analysis, as both show similar karyotypes and recurrent activating JAK1 and STAT3 mutations [33,34,35]. Therefore, clinical assessment of a previous history of breast implants is essential for the correct diagnosis of BI-ALCL and should be combined with the phenotypic characterization, which remains an important tool in the workup of lymphomas [36].

Compared to these previous reports, our study included only specimens of the ALCL subgroup with the ICD-10 code “97023” and expression of CD30 (CD30-postive) and absent tyrosine kinase (ALK-negative). Between 2002 and 2018, twelve cases could be identified in a region under coverage by our center of about 2.1 million inhabitants. With a detection of over 90% of all malignancies diagnosed each year, this allows for a narrowing of the prevalence of PBL-ALCL and BI-ALCL.

However, a few recent studies using different methodology show distinct results.

Doren et al. described the discrepancy of BI-ALCL and primary ALCL of the breast. They meticulously re-analyzed all reported cases in the U.S. between 1996 and 2015 and found an incidence of 2.03 per 1,000,000 person-years, correctly stating that the incidence calculated by the SEER program and previously published by the FDA represents most ALCLs of the breast in the general population. However, they based their assumption and incidence upon textured implants only [10]. In 49 out of 100 identified cases, the implant texturing status was unknown.

A limitation of our study is the unknown number of breast implants and their characteristics, e.g., shell and texturing status as well as filler material among female residents. Furthermore, the regional surgical technique remains unknown. As to the new General Data Protection Regulation (GDPR) and strict manufacturing policy, it was impossible to determine the exact number of implants used each year or contact the according surgeon. This has previously been reported as an international problem. German plastic surgeons therefore gathered to enforce a mandatory breast implant registry starting from 2020. In 2008, however, de Jong et al. reported an incidence of BI-ALCL of 0.1–0.3/100,000 women with breast implants per year based on “uncertain sales data” [16]. In an update of 2018, the incidence was calculated based on the examination of chest x-rays [15,37,38]. Another limitation of our study is the more rural geographic region in eastern Bavaria, with possibly fewer breast implant placement surgeries. However, simultaneously, there is less migration and a constant population level, backing the sensitivity of our cancer registry.

## 5. Conclusions

In conclusion, no case of a CD30-positive and ALK-negative PBL- or BI-ALC was found in a population of 2.1 million inhabitants during the previous 17 years. A precise search algorithm and clinical workup are necessary for case identification [39]. Further research investigating the pathogenesis as well as the incidence and prevalence of PBL- and BI-ALCL including textured and smooth implants is needed. Careful use of classification models and histopathologic subtypes should be the basis for further studies.

A robust national registry for breast implants with an internationally agreed upon data set is necessary to gain solid scientific knowledge on the association between breast implants and anaplastic large cell lymphoma [40] and limit selection bias as well as epidemiologic clusters.

## Figures and Tables

**Figure 1 jcm-09-01247-f001:**
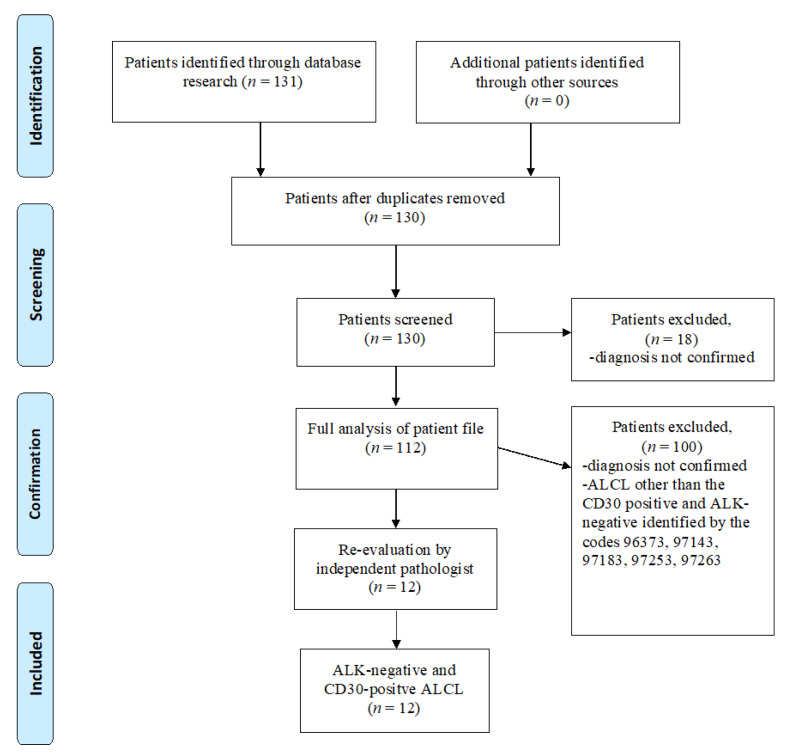
Flow diagram through patient identification, screening, confirmation and inclusion in accordance with the EQUATOR reporting guidelines.

**Figure 2 jcm-09-01247-f002:**
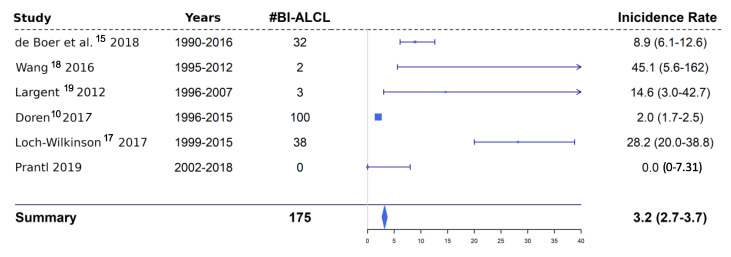
Forest plot showing the incidence rates and confidence intervals of five previous studies in comparison to the Regensburg data. #BI-ALCL = number of breast implant anaplastic large cell lymphomas.

**Table 1 jcm-09-01247-t001:** Annual numbers of ALK-negative and CD-30 positive anaplastic large cell lymphomas (ALCL) in comparison to peripheral T-cell-lymphomas (PTCL) and all other malignancies except for non-melanocytic skin cancer.

Year of Diagnosis	9702/3-ALK-Negativ, CD30-Positive	PTCL: ALK-Positive ALCL, ALCL T and Null Cell Types, ALCL not Specified and without T- and B-Cell Markers, Primary Cutaneous ALCL	C00 to C97 except C44
2002	0	6	8031
2003	0	6	8496
2004	0	7	9909
2005	0	4	9889
2006	0	5	9824
2007	0	6	10,203
2008	0	7	10,427
2009	0	5	10,700
2010	0	7	10,801
2011	1	6	10,369
2012	0	8	10,939
2013	0	9	11,232
2014	2	9	11,131
2015	3	4	11,098
2016	4	5	11,189
2017	1	4	10,705
2018	1	2	5350
Total No. (%)	12 (0.007)	100 (0.05)	170,293 (99.94)

ALK = anaplastic lymphoma kinase, PTCL = peripheral T-cell Lymphoma, ALCL = anaplastic large cell lymphoma, No. = number.

**Table 2 jcm-09-01247-t002:** Organ-based (breast) search algorithm depicting all malignancies found within the breast tissue.

ICD-10	No. of Cases
C82 Follicular lymphoma	4
C83 Non-follicular lymphoma	15
C84 Mature T/NK-cell lymphomas	0
C85 Other and unspecified types of non-Hodgkin lymphoma	2
C50 primary breast cancer	25,897
Total No.	25,918

ICD-10 = 10th revision of international classification of diseases; No. = number.

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
