# Peer review of "Incidence of Anaplastic Large Cell Lymphoma and Breast-Implant-Associated Lymphoma—An Analysis of a Certified Tumor Registry over 17 Years"

_jcm, 2020, doi:10.3390/jcm9051247_

Round 1
Reviewer 1 Report
The present manuscript is a retrospective analysis on the incidence of breast-implant associated ALCL.
In my opinion the analysis is too extended and not easily readable
I would suggest to short the whole article and especially the results and discussion
Many information which is not singificant should be removed in order the article to be more friendly to clinicians and present the main results of the study.
Author Response
Dear Editor, dear Reviewer,
First we would like to thank the reviewer for the more than helpful and critical comments. We really appreciate the time and effort he spent on giving us the necessary criticism for improvement of our manuscript. We did our best to fulfill all suggestions made by the reviewer. We hope to meet all expectations and recommendations in order to have our revised manuscript accepted by the Journal of Clinical Medicine.
In the following paragraphs, we would like to take the opportunity to address all comments by the reviewer in a point-by-point manner (the changes we made to the manuscript are marked as “Track Changes” in the revised version):
Reviewer 1:
In my opinion the analysis is too extended and not easily readable
I would suggest to short the whole article and especially the results and discussion
Many information which is not singificant should be removed in order the article to be more friendly to clinicians and present the main results of the study
Author reply:
We appreciate this comment and agree with the reviewer. Some aspects of our results are redundant and we shortened them accordingly. The discussion deals with some aspects not found within our results (e.g. risk factors) and we shortened them as well.
Reviewer 2 Report
The intent of the study was to established the number/ incidence of ALK- lymphoma in the breast and BreastImplant ALK- lymphoma using the registry of the Bavarian region including 2.1 million inhabitants in Germany..
They found 12 ALK-CD30+ lymphoma but not localized to the breast, 6 were male and 0 BI ALK- CD30+.
They conclude that the incidence in Bavaria was 0 per 1 million breast implant year.
Then, a long comparison with other registries was made showing a higher incidence of BI ALK in other registries or countries. With an average of 3.2
Comments/
The updated WHO classification defined clearly what is BI implant and should be used. It is a sub type of T cell lymphoma. We don’t have data on the incidence of PTCL in your registry.
Biology is still under study but is described by pathologists.
We don’t know also what is the incidence of lymphoma. B cell lymphoma localized to the breast have been reported with a poor prognosis.
Several large reviews have been published on BI and relation with texture implant has been recognized as a potential factor by authorities, EMA and FDA. Although wide variation was recognized according also to regional/country surgical practice.
The discussion is not related to the finding of the study. No case was found in your restricted registry. Regional surgical practice of breast implant is not presented.
Several nationwide registries have been developed and are collecting data for safety.
Other comments:
Table 1; a column showing the number of PTCL or Lymphoma in the registry is warranted. Remove the last column. Use the WHO classification for Lymphoma
Line 142: provide staging with IPI and could be included in Table 2.1
Line 157: ALCL other than ALCL- , please use the who classification
Author Response
Dear Editor, dear Reviewer,
First we would like to thank the reviewer for the more than helpful and critical comments. We really appreciate the time and effort he spent on giving us the necessary criticism for improvement of our manuscript. We did our best to fulfill all suggestions made by the reviewer. We hope to meet all expectations and recommendations in order to have our revised manuscript accepted by the Journal of Clinical Medicine.
In the following paragraphs, we would like to take the opportunity to address all comments by the reviewer in a point-by-point manner (the changes we made to the manuscript are marked as “Track Changes” in the revised version):
Reviewer 2:
The updated WHO classification defined clearly what is BI implant and should be used. It is a sub type of T cell lymphoma.
Author reply:
Thank you for this notation on the current situation regarding the classification of the BI-ALCL. We agree with the reviewer, on the clear definition and classification of the BI-ALCL in the recent classification updates. This a great success led by the understanding and ongoing research about this disease. However, as to the WHO´s lymphoma classification,ALK-negative ALCL was first included as a provisional entity in the 2008 edition by the code “9702/3”. In the third edition before 2008, ALK-negative and ALK-positive lymphomas were listed together as the same entity and were indistinguishable by their classification code alone. Thereafter, BI-ALCL was listed as a provisional entity with the same classification code as ALK-negative and CD30 positive ALCL (“9715/3”). Therefore, the classification and definition has changed over time and mistakes may have been made trying to identify this pathology in databank research. These changes also apply to the ICD-0 as explained in our manuscript. Unfortunately, we cannot change our definition and classification of the BI-ALCL as this is a retrospective databank analysis and trying to convert the classification would lead to serious misinterpretation.
Reviewer 2:
We don’t have data on the incidence of PTCL in your registry.
Author reply:
Thank you for this comment on "PTCL". The "PTCL" describes a broad number of lymphomas with various classification codes and definitions. Apart from the ALK-negative and CD-30 positive ALCL, we did not further investigate these entities and their incidence, as these lymphomas were not of interest for our study. We therefore are unable to comment on the incidence. However, we agree with the reviewer, that the term "PTCL" is more accurate in describing these lymphomas compared to the terms we used in our manuscript. We changed our manuscript accordingly.
Reviewer 2:
“Biology is still under study but is described by pathologists.
We don’t know also what the incidence of lymphoma is. B cell lymphoma localized to the breast have been reported with a poor prognosis.
Several large reviews have been published on BI and relation with texture implant has been recognized as a potential factor by authorities, EMA and FDA. Although wide variation was recognized according also to regional/country surgical practice.”
“Regional surgical practice of breast implant is not presented.”
Author reply:
Thank you for commenting on this serious matter regarding possible factors influencing the incidence of BI-ALCL We agree with the reviewer that various risk factors including texturing status of the implantand bacterial contamination have been under debate. We agree, that a meticulously performed surgical practice with a limited to low amount of intra- and postoperative bleeding/seroma and bacterial invasion is utterly important to significantly reduce the risk for capsular fibrosis and the incidence of BI-ALCL. Regardless, we believe BI-ALCL occurrence has a multifactorial pathogenesis and the aforementioned alone is not the only factor that increases the risk of BI-ALCL. Additional risk factors could potentially be:
- Bacteria as a cause of lymphoma via inflammatory stimulation of lymphocytes
- The role of bacterial super antigens in activating T cell receptors
- High surface area implants are significantly associated with BI-ALCL / surface area is a determinant for higher growth of bacteria in patients and higher stimulation of lymphocytes
- Detection of a Gram-negative microbiome in BIA-ALCL
However,as to the new General Data Protection Regulation (GDPR) and strict manufacturing policy, it was impossible to determine the exact number of implants used each year or comment on any of the aforementioned risk factors. We added this to our limitations section.
Reviewer 2:
“No case was found in your restricted registry”
“The discussion is not related to the finding of the study.”
Author reply:
We acknowledge the fact that no case of ALK-negative and CD-30 positive ALCL was found within the breast tissue and no association to breast implants was detected in our registry. As to the long investigation period and rather larger geographic area, this remains a very significant finding. Equally important remains the workup of these rare lymphomas and why other authors may or may not have found more or fewer cases than we did. This is the main body of our discussion explaining some of the basic rules of these classification systems. However, we agree with the reviewer, that the discussion is too long and deals with topics (e.g. risk factors) that are not in context of our results. We therefore, shortened our discussion for a more precise understanding.
Reviewer 2:
Several nationwide registries have been developed and are collecting data for safety.
Author reply:
We agree with the reviewer on the registry policy. Unfortunately, there is no mandatory breast implant registry in Germany to report postoperative complication. This is one weakness of our study. Since 2019 an official registry is available, which is to become mandatory soon. We therefore might be able to comment on the development of BI-ALCL in the future.
Reviewer 2:
Other comments:
Table 1; a column showing the number of PTCL or Lymphoma in the registry is warranted. Remove the last column. Use the WHO classification for Lymphoma
Author reply:
Thank you for this comment. We subsumed the peripheral T-cell lymphomas except for the ALK-negative and CD-30 positive under “PTCL” in Table 1. Fortunately, the classification of these PTCL was equal in the WHO and ICD-O-3 at the time of classification except for the “97623, ALCL not specified”. We changed our manuscript accordingly and added the nomenclature.
Reviewer 2:
Line 142: provide staging with IPI and could be included in Table 2.1
Author reply:
We agree with the reviewer, that staging is an important tool in oncology. Of course, all patients received full staging and information necessary for the IPI are possibly available. As to the retrospective nature of this study, we fear inaccuracies on giving information about the staging details. We therefore meticulously focused on the clinical information needed, to verify implant status and lymphoma localization.
Reviewer 2:
Line 157: ALCL other than ALCL- , please use the who classification
Author reply:
As explained above, the classification of these PTCL was equal in the WHO and ICD-O-3 at the time of classification except for the “97623, ALCL not specified”. However, we agree with the reviewer that our definition is not accurate and we changed our manuscript adding the nomenclature.
Round 2
Reviewer 1 Report
Manuscript has been improved significantly and it could be published in its present form.
Author Response
Thank you very much. We appreciate your comments.
Reviewer 2 Report
The authors responded to some of the concerns. The manuscript has been improved. The main problem is the quality of the registry where you could not get a clear description of lymphoma of the breast in your region.
Author Response
Dear Editor, dear Reviewer,
First we would like to thank the reviewer for the helpful comments. We really appreciate the critical demand on the methodology of our cancer registry and description/classification of breast lymphomas.
We did our best to fulfill the suggestions made by the reviewer and explain the quality of our registry and further describe the identified breast lymphomas. We now hope to meet all expectations and recommendations in order to have our revised manuscript accepted by the Journal of Clinical Medicine.
In the following paragraphs, we would like to take the opportunity to address all comments by the reviewer in a point-by-point manner (the changes we made to the manuscript are marked as “Track Changes” in the revised version):
Reviewer 2:
The authors responded to some of the concerns. The manuscript has been improved. The main problem is the quality of the registry where you could not get a clear description of lymphoma of the breast in your region.
Author reply:
First we would like to address the concerns regarding the quality of our registry and thereby explain parts of our methodology: The cancer registry receives medical information from all pathologists and clinicians regarding dates of diagnosis, treatment, and follow-up care according to German Cancer Registry Laws. Regarding the coding process and classification (merging ICD-10 codes to diagnosis in registry)we can assure that pathologic coding is solely done on the basis of the original pathologic reports supplemented with results from physician and other treating/diagnostic facilities. Therefore, there is no re-coding but a simple transfer to the registry (by name and ICD-10- so even if some pathologists would have used the WHO or ICD-O-3 classification, we still would find it by name! (spelling mistakes are corrected by the registry)). The description/classification of the identified lymphomas was not made by the authors or the registry. The authors task was to identify these lymphomas and make sure we don’t miss any (because of the heterogeneous/numerous classifications available over a time period of almost two decades) and secure the past medical history. After identification, the pathologic diagnosis was again confirmed by secondary re-analysis of the pathology specimen by an independent pathologist. Therefore, we double-checked the work of the registry and primary pathologist.
Regarding the past medical history of all patients with an anaplastic large cell lymphoma (including ICD-10: 96373, 97143, 97183, 97253, 97263) there has been a full patient analyses including previous surgery and breast implants. All other patients with a NHL other than the described (e.g. follicular lymphoma) during our investigation period have not been further investigated as these were not of interest and there were >5000 cases.
However, as other breast lymphomas and their description seem of particular interest to the reviewer we extended our research and added an organ-based (breast) research approach:
We now added other primary and secondary breast lymphomas as well as primary breast cancer to our manuscript and provided further details on the previous discussed “PTCL´s”. Thereby, ALL malignancies found within the breast are now presented and known from our previous workup, we can assure that no case of ALCL was part of these malignancies.
Among 5181 Non-Hodgkin-Lymphomas (NHL) diagnosed between 2002 and 2018, 21 patients proved to have a localization in the breast (ICD-O-3 C50). None of these 21 patients showed a T-cell-lymphoma and all suffered from B-cell-lymphomas (C82 Follicular lymphoma N=4, C83 Non-follicular lymphoma N=15, C85.9 Non-Hodgkin lymphoma, unspecified N=2).
Therefore, the description and classification of breast lymphomas/malignancies in our region has been precisely outlined.
As to the individual workup, every patient had numerous standardized documents available, including their past medical history and at least one surgical intervention document (biopsy). This goes along with the anesthetic workup of the patient, where previous surgery is mandatory to declare. In our experience, patients tend to honestly declare their past medical history when planning anesthesia on themselves. Furthermore, every patient had multiple CT-scans which were all screened for implants and abnormalities on the chest area, without finding any. Moreover, if treatments in regard to the patient´s disease were performed outside our university (e.g. surgery, chemotherapy, reference histology, recovery, remission), we receive copies even from outside our geographic region, based on German registry laws. If additional information was needed that was not available prior beginning a treatment (past chemo, previous anesthetic workup including surgery to plan a biopsy) these were requested by the treating physician and added by the registry. At last, to receive full assurance, the included patients were phoned and questioned regarding their past medical history including previous surgery with implants.